# Using Cumulus Cell Biopsy as a Non-Invasive Tool to Access the Quality of Bovine Oocytes: How Informative Are They?

**DOI:** 10.3390/ani12223113

**Published:** 2022-11-11

**Authors:** José Felipe Warmling Sprícigo, Ana Luiza Silva Guimarães, Andrielle Thainar Mendes Cunha, Ligiane de Oliveira Leme, Marcos Coura Carneiro, Maurício Machaim Franco, Margot Alves Nunes Dode

**Affiliations:** 1Escola de Veterinária e Zootecnia, Universidade Federal de Goiás, UFG, Goiânia 74690-900, Brazil; 2Departamento de Medicina Veterinária, Centro Universitário Luterano de Palmas, CEULP, Palmas 77019-900, Brazil; 3Centro Universitário de Desenvolvimento do Centro Oeste, Luziânia 72852-580, Brazil; 4Programa de Pós-Graduação em Biologia Animal, Universidade de Brasília, Brasília 70910-900, Brazil; 5Embrapa Recursos Genéticos e Biotecnologia, Laboratório de Reprodução Animal, Brasília 70770-917, Brazil; 6Institute of Genetics and Biochemistry, Federal University of Uberlândia, Uberlândia 38405-32, Brazil; 7School of Veterinary Medicine, Federal University of Uberlândia, Uberlândia 38410-337, Brazil

**Keywords:** cattle, IVM, gene expression, individual culture, embryo

## Abstract

**Simple Summary:**

Assisted reproductive techniques (ART) are used to enhance herds’ genetic gain or to clinically mitigate reproductive failure. Among several options, in vitro embryo production (IVP) allows an efficient dissemination of female germplasm, based on the high number of oocytes available in the ovary. Despite recent progress, many retrieved oocytes are not fully capable to undergo in vitro maturation, fertilization, and culture, resulting in blastocyst development failure. The prediction of oocyte competence is a goal for many research groups on different species. To date, the most promising option to measure the oocyte competence would be evaluating the transcript population of their neighbor cells: the cumulus cells at a transcriptional level. These cells are important mediators of essentials signals and substrates for oocyte to acquire its competence. However, besides many potential candidate’s genes described in the literature, there is no repeatability among different research studies. Moreover, it is not clear if cumulus cell biopsy should be performed on immature or on matured cumulus cells. The present study focused on the evaluation of the potential to predict the oocyte fait after in vitro fertilization, measuring the transcript abundance of a panel of candidate genes on immature and/or mature cumulus cells. The results showed that from all the genes evaluated, none of the then can accurately predict oocyte quality in terms of its potential to develop into an embryo.

**Abstract:**

The present study aimed to determine whether cumulus cells (CC) biopsy, acquired before or after in vitro maturation (IVM), presents similar gene expression pattern and if would compromises oocyte quality. First, immature cumulus oocyte complexes (COCs) were distributed: (1) maturated in groups (control); (2) individually maturated, but not biopsied; (3) subjected to CC biopsy before maturation and individually matured; (4) individually matured and submitted to CC biopsy after maturation; (5) individually matured and CC biopsied before and after maturation. Secondly, candidate genes, described as potential markers of COCs quality, were quantified by RT-qPCR in CCs before and after IVM. After in vitro fertilization (IVF), zygotes were tracked and sorted regarding their developmental potential: fully developed to embryo, cleaved and arrested, and not-cleaved. The COC’s biopsy negatively affects embryo development (*p* < 0.05), blastocyst cell number (*p* < 0.05), and apoptotic cell ratio (*p* < 0.05), both before and after IVM. The PTGS2, LUM, ALCAM, FSHR, PGR, SERPINE2, HAS2, and PDRX3 genes were differentially expressed (*p* < 0.05) on matured CCs. Only PGR gene (*p* = 0.04) was under-expressed on matured CCs on Not-Cleaved group. The SERPINE2 gene was overexpressed (*p* = 0.01) in the Cleaved group on immature CCs. In summary, none of the selected gene studies can accurately predict COC’s fate after fertilization.

## 1. Introduction

It is well established that only competent oocytes can resume and complete maturation, be fertilized, and support the initial embryonic development. Therefore, the efficiency of any in vitro embryo production (IVP) system relies on the availability of competent oocytes [1,2,3,4]. Oocyte competence is achieved gradually and involves nuclear and cytoplasmic modifications, reorganizations of organelles, and intense synthesis and accumulation of RNA and proteins. Therefore, the morphological selection of cumulus–oocyte complexes (COCs) for competent oocytes is limited, which has led to a continuous search for more precise criteria that allow a more accurate selection. Identification of a noninvasive marker for oocyte selection that would improve embryo production, has an undeniable value for ARTs in human. In animals the possibility of selecting more accurately the competent and better-quality oocytes is also of great value especially in situations in which oocytes will be used for cloning and edited embryos, or for producing embryos using rare semen from animals that have already died.

Different approaches have been designed to estimate oocyte quality and predict its developmental fate after fertilization. Oocytes themselves can be used to predict their competence. For example, oocyte diameters and/or sizes of the follicle from which they were derived can be related to oocyte competence and used as predictors [5,6]; however, none of these measurements have proven to be reliable markers. Conversely, investigating molecular markers in the oocytes themselves would be the best way to predict their developmental ability [7]; yet this approach is unsuitable as it prevents subsequent use of oocytes.

Since COC metabolism-derived molecules and products accumulate in the follicular environment [8,9], it can be a more feasible alternative to investigate molecular markers for oocyte competence. Even though follicular fluid has been used in this context in assisted reproductive techniques (ARTs) for humans, its use in predicting COC competence is limited to farm animals. This is mainly because follicular aspiration must be performed individually, which is unfeasible in routine IVP in animals. Another option would be to use follicular cells such as mural granulosa cells (MGCs) and cumulus cells (CCs). MGCs are attached to the follicular wall and are important for hormone synthesis. Matoba et al. [10] and Nivet et al. [11] used MGCs obtained after follicle dissection and found several genes associated with oocyte competence. However, a problem that arises from the use of MGCs, just as with follicular fluid, aspiration should be individual, in addition to the additional procedures to isolate them. That said, the remaining alternative is CCs, which are metabolically coupled with oocytes; therefore, metabolites such as amino acids, saccharides, and signaling molecules, which are all essential for oocyte growth and development [12,13], reach the ooplasm. Such a bidirectional communication with the follicular environment is critical for oocytes, as CC removal or gap-junction blockage before in vitro maturation significantly reduces their capacity to undergo embryonic development [14,15,16]. Given this close relationship, CC can provide important information regarding oocyte health and/or its physiological status [4,17,18]. A variety of studies have been performed in the last two decades, especially in humans, aiming to identify a molecular marker for oocyte quality. Many authors have suggested that transcript levels of candidate genes in human CC biopsies can be associated with oocyte maturation [19,20,21], embryo competence [21,22,23,24,25], pregnancy [17,21,23,26], or live-birth outcomes [21,27,28]. Although a wide variety of genes have been identified and achieved promising results, there is still no consensus on which candidate gene can be considered a reliable marker of oocyte competence in humans or animals [29]. The lack of consistency in results may be due to several factors such as ovarian stimulation protocols, patient characteristics, maturation systems, and oocyte maturational stage.

In ARTs for humans, oocyte retrieval essentially occurs after in vivo maturation after hormonal ovarian stimulation [30,31,32]. Therefore, most of the candidate genes have been identified in CC matured in vivo. On the other hand, in animal models, oocytes for ARTs usually are removed prematurely from non-dominant follicles, and candidate markers have been identified in immature CCs. Moreover, if CCs are obtained from in vitro matured oocytes, such maturation must be considered a different condition so CCs may exhibit significant differences in gene expression than those from matured in vivo oocytes. However, in addition to being identified, a molecular marker must be validated to ensure its reliability in different situations and/or conditions. To do so, CC biopsies must be safely removed to ensure sufficient material, and a single oocyte/embryo culture system must allow the identification of structures until the blastocyst stage.

Given the above, this study aimed to validate the expression patterns of previously described CC candidate genes and correlate them with oocyte competence to develop to the blastocyst stage. First, we tested whether CC removal before and after maturation would not affect embryo development and provided enough material to be used in gene expression analysis. Afterward, we investigated the efficacy of the gene expression of specific candidate genes in CCs acquired before and after in vitro maturation (IVM) to be correlated with oocyte fate after fertilization and culture.

## 2. Materials and Methods

The reagents used were purchased from Sigma Aldrich (St. Louis, MO, USA) unless otherwise stated.

All the COCs used in the present study were aspirated from slaughterhouse ovaries and all the procedures were performed in accordance with the Brazilian Law for Animal Protection.

### 2.1. Experimental Design

#### 2.1.1. Experiment 1: Effects of CC Biopsy and Individual Culture System on In Vitro Embryo Production

This experiment aimed to evaluate whether the biopsy performed in CCs of bovine COCs before and/or after maturation would affect in vitro blastocyst development and its quality. Only COCs with homogeneous cytoplasm surrounded by three or more unexpanded CC layers were selected for in vitro maturation. Biopsies were collected using a scalpel blade and an 8.0 × 0.30 mm insulin syringe (Becton-Dickinson Ultra Fine™, Franklin Lakes, NJ, USA). The experimental groups are described below:Control: The only group wherein in vitro matured (IVM), in vitro fertilized (IVF), and in vitro cultured (IVC) were performed with grouped COCs;Individual IVP: Individual COCs were in vitro matured, fertilized, and cultured;Biopsy before IVM: Immature COCs were subjected to CC biopsy and then individually maturated, fertilized, and cultured in vitro;Biopsy after IVM: Individually matured COCs were subjected to CC biopsy and then fertilized and cultured in vitro;Two Biopsies: Individually matured COCs were subjected to CC biopsies before and after IVM and then followed by in vitro fertilization and culture.

After performing biopsies, the removed CCs were washed and transferred to 0.2-µL tubes. They were then immediately subjected to imaging flow cytometry by FlowSight™ (AMNIS, Seattle, WA, USA) to count the total number of cells. After, cleavage at D2 and blastocyst development at D7 were assessed. To evaluate the effect of CC biopsy on embryo quality, at the end of culturing (D7), developed blastocyst rates were recorded. Moreover, embryos at the blastocyst stage (Bx) were evaluated for total and DNA-fragmented cells by in situ detection of fragmented DNA (TUNEL assay).

#### 2.1.2. Experiment 2: Quantification of mRNA Levels in Biopsies of Immature and Matured Bovine CCs as a Predictor of COC’s Ability to Support Embryo Development

This experiment aimed to identify transcript levels of candidate markers for oocyte competence in bovine immature and matured CCs as a function of their ability to develop into an embryo in vitro.

Since in Experiment 1, no effect of two biopsies was detected on embryonic development, we used them to evaluate gene expression. Therefore, each COC was subjected to biopsies before and after IVM and then individually cultured through all steps of IVP. These biopsied CCs were used for gene expression analysis. A total of 10 replicates were performed, using 625 COCs, which were used for biopsy before and after maturation.

After the biopsy, CCs collected were washed in PBS and transferred to 0.2-µL tubes with RNAlater™ (Ambion™ Life Technologies, Carlsbad, CA, USA) and immediately stored at −20 °C. Cleavage was assessed at D2 and blastocyst development at D7. For gene expression analysis, biopsies were pooled according to COC developmental competence and were classified as: (1) CCs from COCs that developed to blastocyst (EMBRYO); (2) CCs from COCs that cleaved after IVF (D2) but did not reach the blastocyst stage (CLEAVED); and (3) COCs that did not cleave after IVF (NOT-CLEAVED). Then, three pools containing 14 biopsies (each from one COC) were formed for each of the groups.

All selected genes are described in the literature as potential markers of oocyte quality: Glypican-4 (GPC4) [4], Prostaglandin-endoperoxide Synthase 2 (PTGS2) [21,33,34], Activated Leukocyte Cell Adhesion Molecule (ALCAM) [35], Follicle Stimulating Hormone Receptor (FSHR) [5], Progesterone Receptor (PGR) [36], Serine Proteinase Inhibitor Clade E Member 2 (SERPINE2) [37], and Hyaluronic Acid Synthetase-2 (Has2) [38], as well as potential markers of oocyte development: Lumican (LUM) [18], Glutathione Peroxidase 3 (GPx-3) [39], and Peroxiredoxin 3 (PRDX3) [40].

### 2.2. Oocyte Recovery and IVM

Ovaries (*Bos indicus*) were collected immediately after slaughter and transported to the laboratory in saline solution (0.9% NaCl) supplemented with penicillin G (100 IU/mL) and streptomycin sulfate (100 g/mL) at 35 °C. Approximately 1000 COCs were used for all the experiments. Cumulus oocyte complexes (COCs) were aspirated from 3 to 8 mm diameter follicles using an 18-gauge needle and pooled into a 15 mL conical tube. The COCs were recovered and selected in follicular fluid. Only COCs with homogenous cytoplasm and at least three CC layers were used in the experiments. Immediately after selection, COCs were transferred to 150 µL drop of IVM basic maturation media comprising tissue culture media-199 supplemented with 10% fetal calf serum, 0.01 IU/mL porcine FSH (pFSH), 12 IU/mL, L-glutamine, 0.075 mg/mL amikacin, and 0.1 μM cysteamine and were cultured for 22 h at 38.5 °C and 5% CO_2_.

Individual IVM culture was performed in 20 µL microdroplets of the same IVM basic medium described by Kussano et al. [4]. Briefly, in a 60 mm Petri dish, 1620 μL microdroplets were prepared and each drop held an individual COC. The IVM conditions were the same as those described for IVM in the in-group culture.

### 2.3. CC Biopsies

Biopsies of CCs from immature and matured COCs were performed as described by Bunel et al. [18]. The biopsies were conducted individually. The COC was placed in 50-μL drops of follicular fluid previously centrifuged at 700× *g* for 5 min. A very small CC biopsy was removed with an ophthalmic blade (15° Straight; ACCUTOME, Malvern, PA, USA) and an 8.0 × 0.30 mm insulin syringe (Becton-Dickinson Ultra Fine™, NJ, USA). The time to perform the biopsy was approximately of 30 s for each COC. Biopsied immature and matured COCs were washed in PBS and transferred to an IVM or IVF medium, as previously described. Both biopsies were washed in PBS and stored separately in RNA later (Ambion™ Life Technologies, Carlsbad, CA, USA) at −20 °C until gene expression analyses.

Biopsies of bovine CCs removed from immature and matured oocytes were assessed by cytometry. The samples were analyzed using FlowSight Imaging Flow Cytometer (Amnis Corporation, Seattle, WA, USA) equipped with Amnis INSPIRE software (https://www.merckmillipore.com/, accessed on 1 March 2021). For analysis, a single cell from each group/moment was quantified. For single cell acquisition, a specific template was created for cumulus cells, then only events containing cells with size, shape, and positive fluorescence for Hoechst 33342 dye (bisBenzimide H33342) were assessed. H33342 staining was used according to Hallap et al. [41] to exclude potential cellular debris. To do this, cells were previously incubated for 15 min in a buffer solution containing H33342 (0.01 mg/mL), with Hoechst 33342 emissions being collected after exposure to a 405 nm laser at 30 mW. Then, the cell concentration of each sample was subsequently analyzed by the IDEAS V5.0 software (https://ideas.com/, accessed on 1 March 2021).

### 2.4. In Vitro Fertilization and Embryo Culture

Frozen semen from Nellore bull previously tested was used in IVF. Motile spermatozoa were obtained using the Percoll (GE Healthcare, Piscataway, NJ, USA) gradient method in microtubes [42] and were added to the fertilization drop at a final concentration of 1 × 10^6^ spermatozoa/mL. Spermatozoa and oocytes were co-incubated for 18 h at 38.8 °C with 5% CO_2_ in the air. The fertilization medium consisted of Tyrode’s albumin lactate pyruvate (TALP) adapted medium (EMBRAPA, Brasilia, Brazil [43] supplemented with 2 mM penicillamine, 1 mM hypotaurine, 250 mM epinephrine, and 10 mg/mL heparin. The day of in vitro insemination was considered day 0. For individual IVF, a microdroplet system (20 μL) was used. First, a volume of 350 μL fertilization medium was prepared, as described above. Thereafter, the selected spermatozoa were added at the same concentration. The IVF medium containing the spermatozoa was used to prepare the 20 μL microdroplets. Each droplet held an individual matured COC.

After a co-incubation period, presumptive zygotes were washed and transferred to 200 mL droplets of synthetic oviductal fluid (SOF) medium [44] supplemented with essential and non-essential amino acids, 0.34 mM sodium tricitrate, 2.77 mM myo-inositol, and 5% of FBS (Invitrogen™, Waltham, MA, USA). For the culture of individual zygotes, 16 droplets of the same culture medium (20 μL) described above were mounted in a Petri dish and covered with mineral oil.

Embryos were evaluated on day 2 (48 h post-insemination (pi)) for cleavage, and D7 (168 h pi) for blastocyst development.

### 2.5. Total Cell Number (Hoechst 33342) and Apoptotic Cell Ratio (Terminal Deoxynucleotidyl Transferase dUTP Nick End Labeling (TUNEL))

To determine the total cell number and apoptotic cell ratio, expanded embryos were stained with terminal deoxynucleotidyl transferase dUTP nick end labeling (TUNEL) and Hoechst 33342. Blastocysts were washed in warm PBS (Life Technologies, Waltham, MA, USA) supplemented with polyvinyl pyrrolidone (PVP, Life^®^, Carlsbad, CA, USA) (1 mg/mL) before fixation in 4% paraformaldehyde for 1 h. All incubation steps occurred at room temperature in the dark unless otherwise noted, and embryos were washed in 1 mg/mL of PVP between each incubation step. After washing in 1 mg/mL of PVP, the blastocysts were incubated in 0.5% Triton-X for 60 min. Subsequently, positive (artificial DNA denaturation, TUNEL, and Hoechst 33342 staining) and negative (artificial DNA denaturation, Hoechst staining) controls were incubated with 50 U/mL DNase (Roche, Vilvoorde, Belgium) for 1 h. Blastocysts (except negative controls) were then stained with a TUNEL enzyme–labeling mix (Roche) for 60 min at 37 °C and Hoechst 33342 staining for 10 min. Finally, the blastocysts were washed in PVP, mounted on glass slides, and observed under a fluorescent microscope. For each blastocyst, the individual total cell number (blue nuclei, Hoechst 33342) and the total number of apoptotic cells (green nuclei, TUNEL) were determined.

### 2.6. RT-qPCR

The relative abundance of transcripts for ten target genes (GPC4, PTGS2, FSHR, PGR, HAS2, LUM, ALCAM, GPx-3, SERPINE2, and PRDX3) were quantified by qPCR. The qPCR amplification was performed using a 7500 Fast Real-Time PCR System (Applied Biosystems, Foster City, CA, USA) platform. In total, three biological replicates with 14 CCs biopsies/treatments were used. Total RNA (RNeasy Plus Micro, QIAGEN™, Hilden, Germany) from each pool was used for complementary DNA synthesis using 200 U of Superscript III reverse transcriptase (200 U/1 mL; Invitrogen, Waltham, MA, USA) and 0.5 mg of oligo-dT primer (0.5 mg/mL; Invitrogen™, Waltham, MA, USA) in a final volume of 25 µL. The reactions were performed at 65 °C for 5 min and 42 °C for 52 min, followed by enzyme inactivation at 70 °C for 15 min. The qPCR analysis was performed using Fast SYBR Green Master Mix (Applied Biosystems, https://www.thermofisher.com/, accessed on 1 May 2021). The reactions were optimized to provide the maximum amplification efficiency for each gene (80% and 110%) based on calculations using the relative standard curves in the 7500 software 2.0.3 (Applied Biosystems). Each sample was analyzed as technical triplicates, and the specificity of each PCR product was determined by melting-curve analysis and evaluation of amplicon sizes using agarose gels. Each reaction was performed in a final volume of 25 µL using complementary DNA corresponding to 0.35 biopsy, an average of 2267 and 327 CCs in the immature and matured biopsy, respectively. The PCR cycling conditions were 95 °C for 5 min, followed by 50 cycles of denaturation at 95 °C for 15 s, and then annealing at 60 °C for 30 s.

Nomenclature, primer sequences and concentrations, amplicon sizes, and GenBank access number for each primer pair are listed in Table 1. The expression levels of three reference genes, Glyceraldehyde-3-phosphate dehydrogenase (GAPDH), β-actin (ACTB), and Peptidylprolyl isomerase A (PPIA) were subjected to the genNorm software (https://genorm.cmgg.be/, accessed on 1 May 2021) [45], which indicated that all genes were similarly stable, with GAPDH chosen as the reference gene for data normalization. The relative expression of each gene was calculated using the ΔΔCt method with efficiency correction as described by Pfaffl [46].

### 2.7. Statistical Analyses

For cleavage at day 2 and blastocyst development (Bi, Bl, Bx, and Be) at day 7, analysis of variance (ANOVA) was performed, and Tukey’s test was used for means comparison. For cell concentration in biopsies and total and apoptotic cell numbers in blastocysts, Kruskal–Wallis was applied. Gene expression data were evaluated by Student’s t-test. All statistical analyses were performed using the Prophet software version 5.0, 1997 (https://www.prophet-web.com/support/supported-versions/, accessed on 5 June 2021), or GraphPad Prism 6 (https://www.graphpad.com/, accessed on date 5 June 2021), considering *p* values ≤ 0.05 and ≤0.1 as statistically significant and trend, respectively.

## 3. Results

### 3.1. Experiment 1: Effect of CC Biopsy and Individual Culture System on In Vitro Embryo Production

In this experiment, we evaluated whether the moment at which the biopsy is performed affects embryonic development (Table 2). Embryo production did not show differences in individual culture systems compared to in-group control (*p* > 0.05). Likewise, the time that CC biopsy was performed did not affect (*p* > 0.05) blastocyst cleavage or development. However, an interaction between biopsy and individual culture was observed since all groups of oocytes subjected to individual culture and CC biopsies (Immature Biopsy, Matured Biopsy, and Two Biopsies) had lower cleavage (*p* < 0.05) and blastocyst rates (*p* < 0.05) than did control group.

The concentration of cumulus cells obtained from biopsies of immature COCs (6478.2 cells/mL) was higher (*p* < 0.05) than that of biopsies from matured COCs (934.2 cells/mL).

The number of total cells of D7 expanded blastocyst from the control group was higher (*p* < 0.05) than that of individual IVP and all biopsied groups. Regardless of the moment at which it was performed, biopsy affected the percentage of apoptotic cells since it was higher (*p* < 0.05) in the biopsied groups than in the individual IVP and Control groups (Table 3).

### 3.2. Experiment 2: Quantification of mRNA Levels in Biopsies of Immature and Matured Bovine CCs as A Predictor of COC Ability to Support Embryo Development

In Experiment 2, embryo development showed no differences (*p* > 0.05) in individual culture systems between non-biopsied and biopsied groups (Table 4).

For gene expression analysis, we initially compared the abundance of transcripts in immature and mature cumulus cells (Figure 1). The results demonstrated that the expression of eight out of the ten genes analyzed changed during in vitro maturation. The genes PTGS2, PGR, HAS2, LUM, ALCAM, SERPINE2, and PRDX3 were upregulated after IVM (*p* < 0.01). FSHR was the only gene in which transcript levels decreased during IVM (*p* = 0.01). Finally, GPC4 and GPX3 showed similar relative abundance before and after maturation (*p* > 0.05).

Finally, we evaluated the expression levels of the selected genes in immature (Figure 2) and matured (Figure 3) biopsies and correlated them with COC developmental potential. For immature CCs, only SERPINE2 was overexpressed (*p* = 0.01) in the cleaved group. However, matured CCs showed no difference in the same transcript (*p* > 0.05). The gene PGR overexpressed in the not-cleaved (*p* = 0.04) when compared to the cleaved group, but no difference (*p* > 0.05) was observed with the embryo group. Moreover, the expression of the gene LUM tended (*p* = 0.08) to be lower in matured CCs on COCs from the not-cleaved group when compared to the others.

## 4. Discussion

Oocyte competence is gradually acquired during oogenesis and results from interactions between oocytes and follicular cells [47,48]. Cumulus cells are somatic cells that maintain a metabolic relationship with the oocyte, supporting maturation and competence acquisition [15,49]. Therefore, CCs can give us important information about oocyte status and could be used as a non-invasive tool to select more competent oocytes. Even though a variety of studies have reported candidate genes as molecular markers to predict oocyte competence, there is no consensus about which genes could be used as a marker. Especially in domestic animals, such information is hindered not only due to the fewer studies but also due to differences among species and in vitro systems. Thus, our study proposed to evaluate whether CC biopsy could interfere with embryo development and quality after fertilization. The study also evaluated whether the gene panel already established as markers of human oocyte competence may also work for bovine COCs before and after IVM.

In the first experiment, the effect of biopsies performed before and/or after IVM on embryo development was evaluated. Since CC biopsies were mainly used to detect molecular markers, in the first assay the number of feasible cells to be obtained from immature and mature CC biopsies was quantified. The results showed that biopsies from immature COCs had a higher number of cells compared to biopsies from matured ones. The lower cell numbers in matured COCs may have been due to cumulus expansion. Under such conditions, cells are dispersed in hyaluronic acid, thus, in same-sized biopsies, fewer cells are recovered in matured COCs. Therefore, even with a fewer number of cells, matured COCs had enough material to quantify gene expression by qPCR. Moreover, a housekeeping gene was used to normalize the assay, allowing comparisons between the relative abundances of matured and immature CCs.

Regarding the biopsy effect, our results showed that its performance before and/or after maturation did not impact embryo development. In the first experiment, the individual system alone was unable to negatively affect blastocyst development, but a biopsy and individual culture interaction negatively affected embryo production. In contrast, Bunel et al. [18] found no effect of individual culture on embryo development. Based on our results and in the literature [4], we can assume that individual culture potentialized by biopsy procedure does negatively affect embryo development. Grouped COCs/embryos are widely known to be exposed to autocrine and paracrine molecules produced and secreted by “neighbor” counterparts. Among the factors secreted in the group, the system is insulin-like growth factors I and II (IGF-I, IGF-II), transforming growth factor α and β (TGF-α, TGF-β), interferon τ (IFN-τ), epidermal growth factor (EGF), platelet activated factor (PAF), platelet-derived growth factor (PDGF) [50,51], and glucose and other energy substrates [52]. All these molecules may be important for the regulation of oxidative stress, intercellular communication, and activation of pathways responsible for cell proliferation. Thus, the beneficial effect of a group culture is expected to be lost during individual culture [53,54,55], perhaps due to the impact of bench handling on COC quality.

We also evaluated the quality of embryos. In all individually cultured groups, expanded blastocysts had lower cell counts, suggesting a poorer embryo quality. The total number of cells can estimate the developmental potential of embryos after transfer. Blastocysts with higher cell numbers are more likely to have success during maternal recognition of pregnancy and have reduced pregnancy losses in many species [56,57]. Moreover, total cell counts have a positive correlation with embryo resistance during cryopreservation [58]. Such an observation reinforces the above discussed, paracrine factors produced and secreted are important for proper embryo development. To enhance our analyses on embryo quality, the percentage of apoptotic cells was determined in the same embryos. The results demonstrated that biopsy increased apoptosis rates in the expanded blastocysts analyzed. CC communication is important for proper oocyte maturation; however, performing a biopsy on matured COCs induced apoptosis at the same intensity as in immature ones. This may have occurred due to the metabolic co-dependence between oocytes and CCs, which is important not only during maturation but also during fertilization [52]. In addition, a negative effect of manipulation itself cannot be ruled out, as there may be a degree of invasiveness that affects oocytes. To conclude the first goals, we must assume that individual IVP and CC biopsy affect total cell number and apoptotic rates, respectively. Despite their negative impact on embryo quality, these two procedures are essential to track oocyte cells and produce accurate information about biopsy effect on their further development.

In the second experiment, we aimed to validate candidate genes as potential markers for oocyte competence. Unlike other studies, ours was the first study to use the same COCs to acquire biopsies of CCs, before and after IVM. The use of CC biopsies from the same COCs, in both moments, allows tracking oocytes precisely throughout the entire process. Moreover, based on the results in Experiment 1, no difference was observed in embryo development and its quality, irrespective of the biopsy moment. Therefore, we quantified gene expression on immature and matured CCs and then correlated it to the ability of COCs to develop into embryos.

During maturation, COCs undergo functional, morphological, and molecular transformations to ensure their progression to MII, reorganization of organelles, and expansion of CCs. These, together, confer female gamete competence to be fertilized and develop to later stages after fertilization [7,52]. Data from the last decade show that a dynamic process in the abundance of transcripts in CCs [59] and oocytes [60] occurs to coordinate these aforementioned events. These differences may lead to misinterpretation of the molecular panel of competence genes designed in a study to be used in another one. Moreover, breed [61], age [62], follicle size [63,64], maturation system, and media [65] affect CC molecular signature. Therefore, these factors can reduce COC quality prediction accuracy, based on the expression of the competence gene marker. Therefore, using the same COCs to acquire CC biopsy before and after maturation would exclude all confounding factors and point out the real differences between both time points. In the present study, we found that 7 out of 10 genes were overexpressed after maturation (LUM, PTGS2, ALCAM, PGR, SERPINE2, HAS2, and PDRX3). Moreover, 1 (FSHR) and 2 (GPC4 and GPX3) out of 10 genes were downregulated or stable after IVM, respectively. Such information is important because, in bovines, the acquisition of CCs for molecular analyses is often performed in immature COCs [4,10,60], different from humans [17,19,23,24,25,27,28,66]. Therefore, the gene panel designed for different species may not be the same across all mammals.

The molecular markers of immature COCs have a high correlation with the maturation process. Using the same markers to predict the quality of matured COCs may be inefficient, as the pathway in which they participate will no longer be required after maturation. Therefore, a panel of genes has to be designed for a specific stage. In our study, the gene expression panel was compared among immature and mature CCs, depending on the COC fate. After fertilization, structures that did (CLEAVED) or did not (NOT-CLEAVED) cleave on Day 2, and those that reached the blastocyst stage on D7 (EMBRYO) were used as phenotypes. The gene LUM tended to be overexpressed (*p* = 0.08) in CCs from matured oocytes that reached the blastocyst stage or cleaved when compared to oocytes that did not cleave (not-Cleaved). The gene LUM plays an important role in extracellular matrix regulation, is present in cumulus expansion and oocyte maturation, and is also associated with cell proliferation, migration, apoptosis, and angiogenesis [67]. When evaluating the level of transcripts of the LUM gene in immature and matured oocytes after in vivo and in vitro maturation, Mamo et al. [60] found a progressive increase in the amount of mRNA after meiosis resumption, with a higher number of transcripts after maturation. However, Bunel et al. [18] found that less competent oocytes, which failed to become embryos after IVM, IVF, and IVC, presented high levels of LUM mRNA in immature COCs. Such difference may be because the follicles used to obtain COCs belong to a heterogeneous class of follicles, as they may be in the recruitment or atresia phase, which could influence the degree of competence of oocytes.

Moreover, the gene PGR was the only one to show a significant difference (*p* = 0.04), and a higher expression was observed in CCs from cleaved COCs than from not-cleaved ones. The gene PGR was reported to be induced by LH and suggested as a crucial playmaker (hormone receptor and transcription factor) upstream of the ovulatory process pathway [68]. Therefore, we cannot explain why this gene was overexpressed in CCs from cleaved COCs compared to both extremes, the ones that did develop into embryos or, on the other side, the ones that did not cleave.

Evaluating immature CCs, the gene SERPINE2 showed a similar pattern, with higher expression (*p* = 0.01) in cleaved ones compared to those that did not or become embryos. SERPINE2, also known as protease nexin-1, belongs to the serine protease inhibitor SERPIN superfamily. It is one of the potent SERPINs that modulate the activity of plasminogen activators [69]. For humans, mature CCs with higher expression were associated with higher blastocyst development [70] and pregnancy [37]. Our findings make clear the impossibility of extrapolating information from humans to other mammalian species.

Finally, the present study demonstrated that transcript levels of the genes PTGS2, ALCAM, SERPINE2, HAS2, and PDRX3 are indeed related to COC maturation, corroborating the literature data [28,66,71,72]. However, no correlation was found regarding COC fate after IVF. Finally, analyzing the genes GPC4 and GPX3, no difference was observed either concerning the maturation stage or the ability of COCs to cleave (D2) or to develop into embryos (D7).

A non-invasive method for COC competence assessment should indicate oocytes that are more competent to develop after IVF and IVC. Such differentiation is the most important step to identifying precociously the developmental potential of embryos. However, gene expression varies not only because of maturation status and COC fate, as demonstrated in our results. Still, several other factors beyond our findings may influence COC fate [63,65].

## 5. Conclusions

Based on our results, we concluded that none of the selected genes can accurately predict oocyte quality in terms of its potential to develop into an embryo after fertilization. In addition, because gene expression in cumulus cells varies along maturation, studies using genes expression for oocyte quality investigation must consider whether cumulus cell biopsy will be acquired before or after maturation.

## Figures and Tables

**Figure 1 animals-12-03113-f001:**
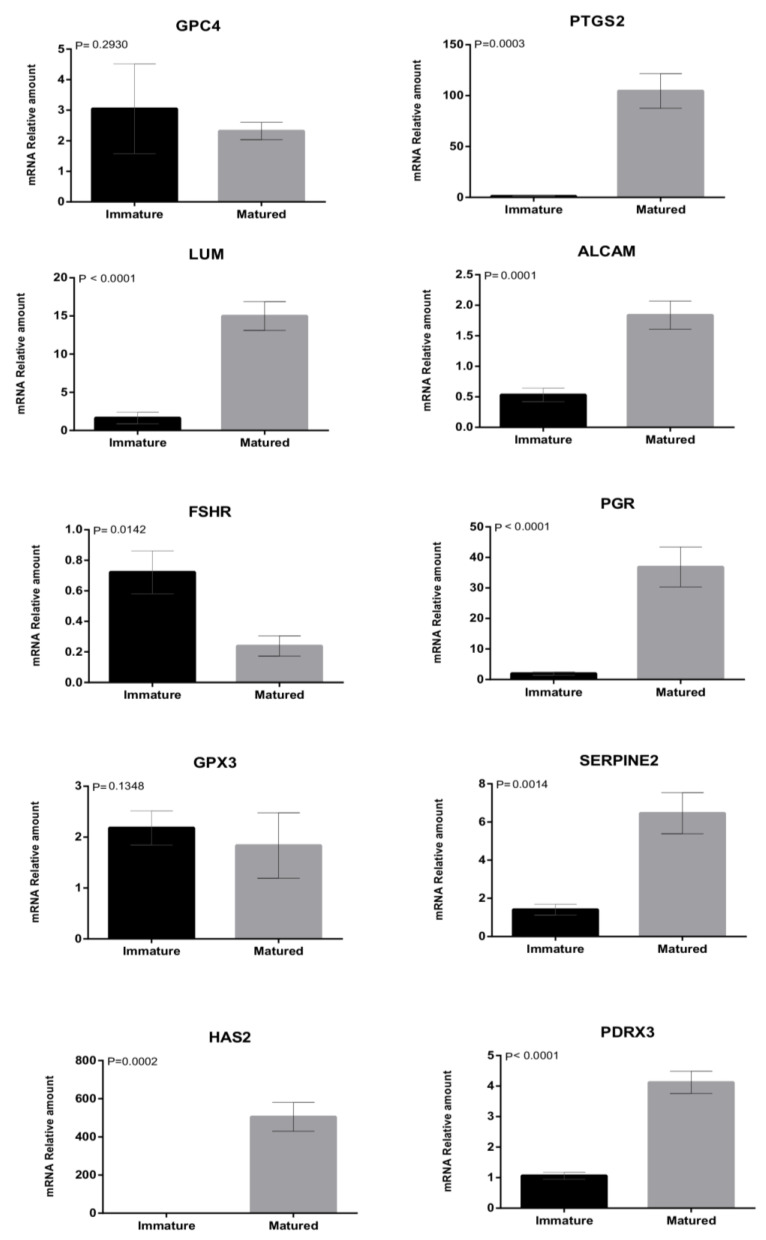
Relative abundance of messenger RNA (mRNA) encoding GPC4, PTGS2, LUM, ALCAM, FSHR, PGR, GPX3, SERPINE2, HAS2, and PRDX3 genes determined by quantitative polymerase chain reaction in immature (black bar) or CCs of COCs undergoing maturation (gray bar). Mean ± standard error of the mean (SEM) of three biological replicates. The data were normalized using the formula DDCT by Pfaffl [46], with GAPDH as an endogenous control. The differences were significant when *p* < 0.05, according to the T test.

**Figure 2 animals-12-03113-f002:**
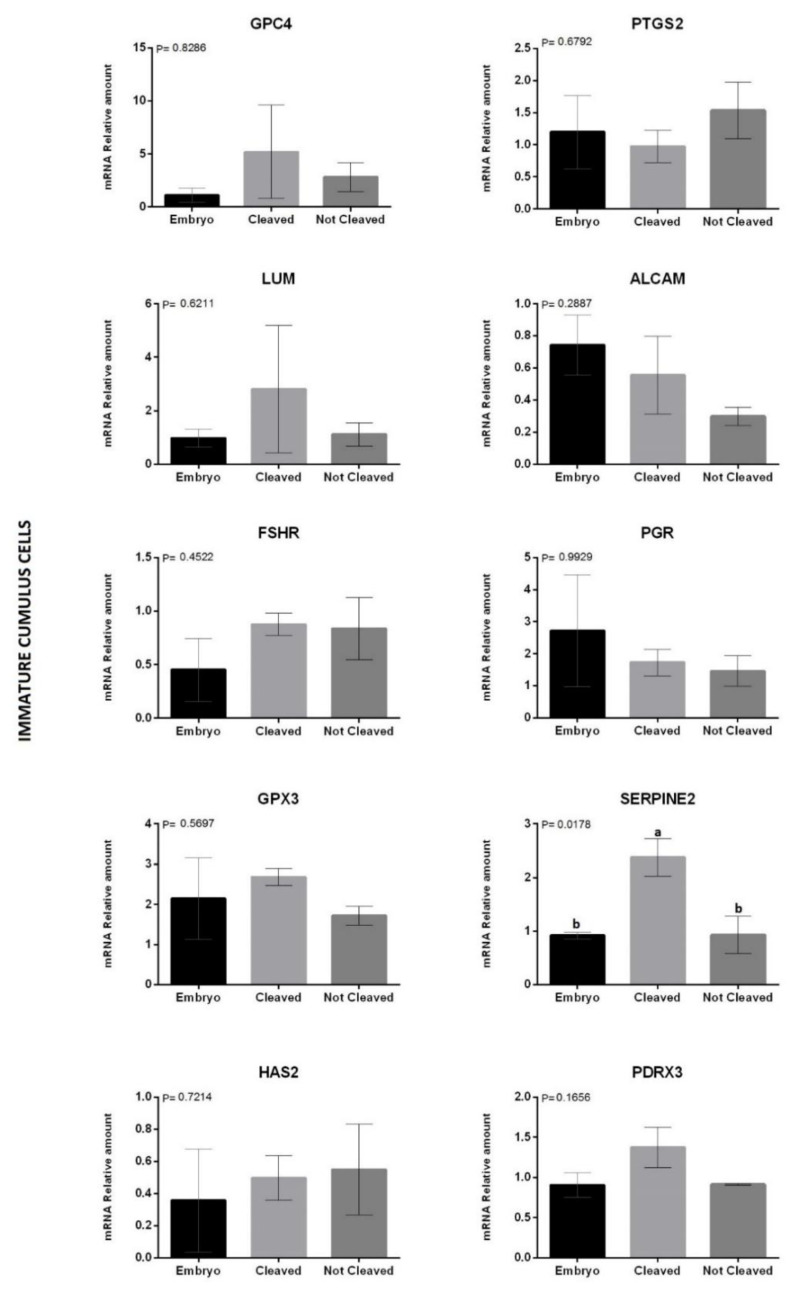
Relative abundance of messenger RNA (mRNA) encoding GPC4, PTGS2, LUM, ALCAM, FSHR, PGR, GPX3, SERPINE2, HAS2, and PRDX3 genes determined by quantitative polymerase chain reaction. Samples acquired on immature CCs biopsy from COCs that developed until embryo (black bar); cleaved but did not develop (bright gray bar) or did not cleave (dark gray bar) after IVM, IVF, and IVC. Mean ± standard error of the mean (SEM) of three biological replicates. The data were normalized using the formula DDCT by Pfaffl [46], with GAPDH as an endogenous control. ^a,b^ Statistically significant differences between treatments. The differences were significant when *p* < 0.05, according to the T test.

**Figure 3 animals-12-03113-f003:**
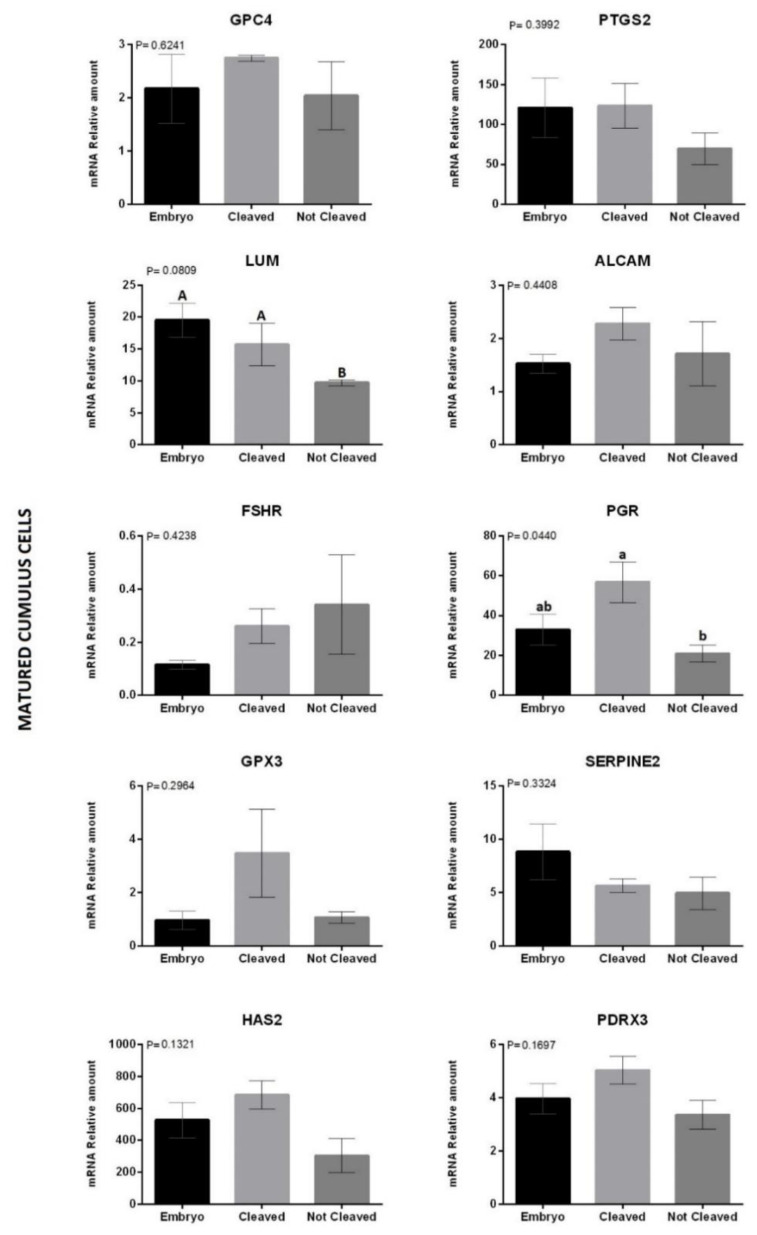
Relative abundance of messenger RNA (mRNA) encoding GPC4, PTGS2, LUM, ALCAM, FSHR, PGR, GPX3, SERPINE2, HAS2, and PRDX3 genes determined by quantitative polymerase chain reaction. Samples acquired on matured CCs biopsy from COCs that developed until embryo (black bar); cleaved but did not develop (bright gray bar) or did not cleave (dark gray bar) after IVM, IVF and IVC. Mean ± standard error of the mean (SEM) of three biological replicates. The data were normalized using the formula DDCT by Pfaffl [46], with GAPDH as an endogenous control. ^a,b^ statistically significant differences and ^A,B^ tendency between treatments The differences were considered to be significant and a tendency, when *p* < 0.05 and *p* < 0.1, respectively, according to the T test.

**Table 1 animals-12-03113-t001:** Information about the specific primers used for the amplification of gene fragments for the quantitative polymerase chain reaction analysis.

Genes	Primer Sequences	Amplicon Size (bp)	Primer Concentration (nM)	GenBank Access Number/Reference
*GAPDH*	F: GGC GTG AAC CAC GAG AAG TAT AA R: CCC TCC ACG ATG CCA AAG T	118	300	NM_001034034.2
*GPC4*	F: TGG TGA ATC CCA CAA CCC AGT GTA R: TCT CAG CCA CCA TCA GCA TAG CAT	192	300	NM_001205784.1
*LUM*	F: GTC TCC CAG TGT CTC TTC TAA R: GAG ATC CAG CTC CAA CAA AG	179	300	NM_173934.1
*PTGS2*	F: GAG GAA CTT ACA GGA GAG AAG R: CGG GAG AGC ATA TAG GAT TAC	193	250	NM_174445.2
*ALCAM*	F: GGA CAG CCT GAA GGA ATT AG R: CCA ATC TGC TTA GTC ACC TC	182	300	NM_174238.1
*FSHR*	F: GGA TGC CAT CAT CGA CTC TG R: TGA CTC GAA GCT TGG TGA GAA C	133	300	NM_174061
*GPX3*	F: GCT AGA CCC TTT ACT GTT ACA C R: GTT CCT CTC TGG CAT TCT TC	189	300	NM_174077.4
*PGR*	F: TCAGGCTGGCATGGTTCTTGG R: CTTAGGGCTTGGCTTTCGTTTGG	126	300	NM_001205356.1
*SERPINE2*	F: GAC TCC TTT CCT ACA TCT TTC C R: CAG TAC AGT GTT CCA CCA TC	158	300	NM_174669.2
*HAS2*	F: GGG TTC TTC CCT TTC TTT CT R: CCA CCC AGC TTT GTT TAT TG	240	250	NM_174079.2
*PDRX3*	F: GGC AGG AAC TTT GAT GAG AT R: GTG TGT AGC GGA GGT ATT TC	205	300	NM_174643.1

F: primer forward; R: primer reverse.

**Table 2 animals-12-03113-t002:** Number (*n*) and percentage ± standard deviation (% ± SD) of cleavage (D2) and embryo development (D7). Cumulus oocyte complexes (COCs) were submitted to in vitro maturation, fertilization, and culture in groups (Control), individually (Individual IVP), or individually submitted to cumulus cells (CC) biopsy, before (Immature Biopsy), after (Matured Biopsy), and before and after in vitro maturation (Two Biopsies).

Treatment	Oocytes *n*	Cleavage at D2 *n (% ± S.D.)*	Blastocyst at D7			
			Bi *n (%)*	Bl *n (%)*	Bx *n (%)*	Total *n (% ± SD)*
Control	177	135 (76.2 ± 5.0) ^a^	19 (28.0%)	25 (37.0%)	24 (35.0%)	68 (38.4 ± 7.8) ^a^
Individual IVP	112	76 (68.0 ± 17.4) ^a,b^	12 (32.4%)	16 (43.2%)	9 (24.4%)	37(33.0 ± 5.1) ^a,b^
Immature Biopsy	112	63 (56.2 ± 8.5) ^b^	9 (29.0%)	16 (52.0%)	6 (19.0%)	31 (27.6 ± 4.2) ^b^
Matured Biopsy	112	70 (62.5 ± 4.5) ^b^	6 (21.6%)	15 (53.5%)	7 (25.0%)	28 (25.0 ± 4.3) ^b^
Two Biopsies	112	67(60.0 ± 4.0) ^b^	13 (45.0%)	9 (31.0%)	7 (24.0%)	29 (25.8 ± 3.5) ^b^

^a,b^ Values with different superscripts in the same column are significantly different by ANOVA (*p* < 0.05). Initial blastocyst (Bi), blastocyst (Bl) and expanded blastocyst (Bx).

**Table 3 animals-12-03113-t003:** Means and standard deviation (mean ± SD) of the number of total cells and the percentage (%) of apoptotic cells of D7 expanded blastocyst (Bx). Embryos were originated from cumulus oocyte complexes (COCs) submitted to in vitro maturation, fertilization, and culture in groups (Control), individually (Individual IVP) or individually and submitted to cumulus cells (CC) biopsy, before (Immature Biopsy), undergoing maturation (Matured Biopsy), and before and after in vitro maturation (Two Biopsies).

Treatments	Total Number of Cells *Mean ± SD*	Apoptotic Cells Ratio *%*
Control	134.0 ± 24.8 ^a^	3.73 ^a^
Individual IVP	118.0 ± 18.1 ^b^	3.81 ^a^
Immature Biopsy	113.0 ± 20.2 ^b^	6.72 ^b^
Matured Biopsy	119.2 ± 19.3 ^b^	5.62 ^b^
Two Biopsies	115.5 ± 19.0 ^b^	7.53 ^b^

^a,b^ Values with different superscripts in the same column are significantly different (*p* < 0.05), according to Kruskal–Wallis test.

**Table 4 animals-12-03113-t004:** Number (*n*) and percentage ± standard deviation (% ± SD) of cleavage (D2) and embryo development (D7). Cumulus oocyte complexes (COCs) individually matured, fertilized, and cultured were submitted or not (Control) to a biopsy before and after in vitro maturation (Two biopsies).

Treatment	Oocytes *n*	Cleavage at D2 *n (% ± S.D.)*	Blastocyst at D7				
			Bi *n (%)*	Bl *n (%)*	Bx *n (%)*	Be *n (%)*	Total *n (% ± SD)*
Control	160	100 (62.5 ± 14.4)	14 (28.5%)	12 (24.5%)	19 (38.7%)	4 (8.1%)	49 (30.6 ± 7.5)
Two Biopsies	160	98 (61.2 ± 21.9)	13 (28.2%)	15 (32.6%)	18 (39.1%)	0 (0.0%)	46 (28.7 ± 9.6)

All data were analyzed by ANOVA. Initial blastocyst (Bi), blastocyst (Bl), expanded blastocyst (Bx), hatching, and hatched blastocyst (Be).

## Data Availability

All available data are included in the present paper.

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
