# Peer review of "Using Cumulus Cell Biopsy as a Non-Invasive Tool to Access the Quality of Bovine Oocytes: How Informative Are They?"

_animals, 2022, doi:10.3390/ani12223113_

Round 1
Reviewer 1 Report
The authors present an interesting, well-written and detailed work. The experimental design and the methodology are clear and the results are well expressed to try to respond to the stated objectives. The conclusions could be more detailed.
Author Response
Conclusions were reformulated as suggested.
Reviewer 2 Report
Comments to the author:
The manuscript “Using Cumulus Cell Biopsy as A Non-invasive Tool to Access the Quality of Bovine Oocytes: How Informative Are They?” presents interesting and important results for the understanding of the developmental competence of bovine oocytes. I suggest authors review the following comments for better publication quality:
In the simple summary and abstract:
1. In the simple summary, I suggest the authors to insert a sentence at the end informing the main result achieved, that is, did the authors answer affirmatively or not to the question that is the title of the manuscript?
2. In “individually matured and CC biopsied”, explain better what group 4 would be.
3. In “fully developed to Embryo, Cleaved and arrested, and Not-Cleaved”, Could this not be replaced by assessment of embryonic development only? In addition, did the authors assess developmental kinetics and embryonic quality?
4. In “The COC’s biopsy negatively affects embryo development (P<0.05), blastocyst cell number (P<0.05), and apoptotic cell ratio (P<0.05)”, both before and after IVM? To specify.
5. In “were differentially expressed (P<0.05) on matured CCs”, I suggest the authors explain this statement further.
6. In the conclusion “the CC biopsy of immature and matured COCs can be used to assess and measure gene expression.”, how could it be used, if it negatively affected the oocytes?
In the introduction:
1. I was wondering about a practical application of the biopsy. Would it be possible, within IVP routines, to perform biopsies before IVM and define which oocytes would go to IVM?
2. Furthermore, wouldn't individualized cultivation be a much more evident reality in humans than in cattle?
In the material and methods:
1. As for the CC biopsy, I suggest that the authors detail the procedure, specifying how much of each COC is removed during the procedure, time of performance, in which medium the biopsy is performed, etc.
2. Inform the composition of the IVM medium used in the present study.
In the results and discussion:
1. In the phase “The concentration of cumulus cells obtained from biopsies of immature COCs (6,478.2 cells/mL) was higher (P<0.05) than that of biopsies from matured COCs (934.2 cells/mL).”, it is not clear what this means. The authors performed the same procedure, check? They collected the same amount, check? Why would this difference in concentration be expected or occurred? And what implications would that have?
2. In “mature cumulus cells” (L319) and figure 1, the most correct would be to say of oocytes undergoing maturation, check? You can't say that it was CCs of COCs that matured.
Author Response
In the simple summary and abstract:
- In the simple summary, I suggest the authors to insert a sentence at the end informing the main result achieved, that is, did the authors answer affirmatively or not to the question that is the title of the manuscript?
In fact, from all the genes evaluated none of the then can accurately predict oocyte quality in terms of its potential to develop into an embryo after fertilization. Therefore, the use of those genes in the conditions that the work was performed gene expression gave no information about oocyte competency. A sentence with that information was added into the simple summary as suggested.
- In “individually matured and CC biopsied”, explain better what group 4 would be.
The text was modified to clarified group 4 (individually matured and submitted to CC biopsied after maturation)
- In “fully developed to Embryo, Cleaved and arrested, and Not-Cleaved”, Could this not be replaced by assessment of embryonic development only? In addition, did the authors assess developmental kinetics and embryonic quality?
We understand the considerations. However, the objective was to explain that based on the zygote faith after fertilization [fully developed to Embryo (i), Cleaved and arrested (ii), and Not Cleaved(iii)] gene expression of selected genes was evaluated. The information of the embryonic quality was accessed by the total number of cells and apoptotic cell ratio, that is presented in the abstract.
- In “The COC’s biopsy negatively affects embryo development (P<0.05), blastocyst cell number (P<0.05), and apoptotic cell ratio (P<0.05)”, both before and after IVM? To specify.
A sentence with that information was added into the abstract as suggested.
- In “were differentially expressed (P<0.05) on matured CCs”, I suggest the authors explain this statement further.
The statement is better explained in the results section. According to the guideline there is a limit of 200 words for the abstract. Therefore, it is not possible to explore all the data beyond already written.
- In the conclusion “the CC biopsy of immature and matured COCs can be used to assess and measure gene expression.”, how could it be used, if it negatively affected the oocytes?
The conclusion was modified as suggested.
In the introduction:
I was wondering about a practical application of the biopsy. Would it be possible, within IVP routines, to perform biopsies before IVM and define which oocytes would go to IVM?
The used of the biopsy for human ARTs is unquestionable, but we understand the reviewer concern about the practical application of the biopsy in animal embryo production. We think that it will have no use in a commercial routine where thousands of eggs from different animals are used every day. However, in certain situations, the possibility of selecting more accurately the competent and better-quality oocytes would be of great value. Situations in which oocytes will be used for cloning and edited embryos, or for producing embryos using rare semen from animals that have already died, it would be worth do a biopsy, a PCR and use only the best oocytes for embryo production.
Furthermore, wouldn't individualized cultivation be a much more evident reality in humans than in cattle?
We agree with the reviewer, but it is important to remember that the individual culture was used in the present study as a tool to follow the oocytes until the blastocyst stage. In fact, if one wants to use the molecular markers to select oocytes, it would be necessary to keep the oocytes individually only a until the PCR result, when the most competent ones can then be chosen and used for IVP. Individual culture may also be necessary, when donors give very few oocytes, which occurs in some breeds and some especial animals.
In the material and methods:
- As for the CC biopsy, I suggest that the authors detail the procedure, specifying how much of each COC is removed during the procedure, time of performance, in which medium the biopsy is performed, etc.
The following information was added to the text as suggested. It is not possible to measure the biopsy size, however, as described it was taken the smallest possible.
“The biopsies were conducted individually. The COC was placed in 50-μL drops of follicular fluid previously centrifuged at 700 × g for 5 minutes. A very small CC biopsy was removed with an ophthalmic blade (15° Straight; ACCUTOME, Malvern, PA, USA) and an 8.0 x 0.3 mm insulin syringe (Becton-Dickinson Ultra Fine™, NJ, USA). The time to perform the biopsy was approximately of 30 seconds for each COC.”
- Inform the composition of the IVM medium used in the present study.
The IVM medium composition was added to the text as suggested.
In the results and discussion:
- In the phase “The concentration of cumulus cells obtained from biopsies of immature COCs (6,478.2 cells/mL) was higher (P<0.05) than that of biopsies from matured COCs (934.2 cells/mL).”, it is not clear what this means. The authors performed the same procedure, check? They collected the same amount, check? Why would this difference in concentration be expected or occurred? And what implications would that have?
Yes, the authors conducted the same procedure. Yes, the authors subjectively collected the same size. However, the information reveals that the cell number is different in the immature and in the matured cumulus cells biopsy. The author’s hypothesis is that due to cumulus expansion during IVM a fragment with similar size has fewer cells, when compared to immature biopsy. The expansion occurs due to the production of hyaluronic acid during the IVM. All these comments are presented in the second paragraph of the DISCUSSION section.
- In “mature cumulus cells” (L319) and figure 1, the most correct would be to say of oocytes undergoing maturation, check? You can't say that it was CCs of COCs that matured.
The sentences were changed as suggested.
Reviewer 3 Report
Main issues: ethical approval for every experimental and appropriate explanation of M&M.
Some other comments:
- ln 35. IVM? First time you use an acronym, please explain it.
- ln 35. and if IT would...
- ln 36. COCs?
- ln 40. IVF?? Take into account that simple summary not always appear with asbtract... So, I would mention this acronym here as well.
- ln 41-42. is bold really necessary?
- ln 44. only PGR "gene"
- Introduction is well written and structured and it provide an interesting state-of-the-art of the topic. However, I missed a more cattle-focused approach instead of showing human results. This change could be useful to show the gap knowledge that justifies this interesting study.
- M&M is not appropriately structured, nor the ethis committe approval appears, which is completely needed and highly important.
- In my opinion the structure would be as follows: 1) animals, IVM and biopsies; 2) IVF and embryo culture; 3) measurements and procedures applied
- ln 125-133. Maybe a figure explaining the groups would be very visual and enhance clarity of the manuscript.
- ln 167. Bos indicus in italic
- ln 170 an so on. Use long -, instead of -
- missing the number of oocytes, ovaries used, number of uterus, etc. Provide this detailed information
- ln 174. use the appropriate degree symbol
- ln 199. just one bull?
- Table 2. The statistical procedure it is not adequate. Repeated measurements over time are needed; you are comparing initial blastocyst vs. blastocyst vs. expanded blastocyst. If you are just measuring the aumount of each at day7 and comparing the stage, yes the statistical procedure is correct, but explain previously clear.
- I would merge Table 3 with Table 2 showing the apoptotic cells ratio.
- Table 4. where are the superscripts? If there are no differences 8as said in ln 309-310), delete this sentence under the table.
- figure 1. PTGS2 and HAS2, missing "=" after p value
- ln 334-339. why are you using capital letters? I think it is not necessary
- figure 3. all superscripts in the same format
Author Response
- Main issues: ethical approval for every experimental and appropriate explanation of M&M.
Answer: All the COCs used in the present study were aspirated from slaughterhouse ovaries and all the procedures were performed in accordance with the Brazilian Law for Animal Protection, which recommends that slaughterhouse material must follow the slaughter legislation established by the Brazilian Ministry of Agriculture (MAPA). A sentence regarding to legislation was added in the M&M.
Some other comments:
- ln 35. IVM? First time you use an acronym, please explain it.
Answer: The sentence was changed as suggested.
- ln 35. and if IT would...
Answer: As suggested the sentence was changed to better comprehension.
- ln 36. COCs?
Answer: The sentence was changed as suggested.
- ln 40. IVF?? Take into account that simple summary not always appear with asbtract... So, I would mention this acronym here as well.
Answer: The sentence was changed as suggested.
- ln 41-42. is bold really necessary?
Answer: The text was changed as suggested.
- ln 44. only PGR "gene"
Answer: The word “gene” was added.
- Introduction is well written and structured and it provide an interesting state-of-the-art of the topic. However, I missed a more cattle-focused approach instead of showing human results. This change could be useful to show the gap knowledge that justifies this interesting study.
Answer: A sentence was added into the text to better justify the importance of the study to animal embryo production.
- M&M is not appropriately structured, nor the ethis committe approval appears, which is completely needed and highly important.
Answer: Animals were not used, all the COCs were obtained from slaughterhouse ovaries. All the COCs used in the present study were aspirated from slaughterhouse ovaries and all the procedures were performed in accordance with the Brazilian Law for Animal Protection, which recommends that slaughterhouse material must follow the slaughter legislation established by the Brazilian Ministry of Agriculture (MAPA). A sentence regarding to legislation was added in the M&M.
- In my opinion the structure would be as follows: 1) animals, IVM and biopsies; 2) IVF and embryo culture; 3) measurements and procedures applied
Answer: We emphasize that we only use slaughterhouse ovaries in the present study. Thus, there was no animal handling for the experiment. We only work with post-mortem material. Therefore, we believe that the presented sequence is in agreement for a proper comprehension.
- ln 125-133. Maybe a figure explaining the groups would be very visual and enhance clarity of the manuscript.
Answer: Thank you for suggesting. The authors agree with the other reviewers that the given explanation is well structured and explanatory. However, if the reviewer thinks that the figure is mandatory for a better experimental design comprehension, we will add it.
- ln 167. Bos indicus in italic
Answer: The text was changed as suggested.
- ln 170 an so on. Use long -, instead of -
Answer: Changed as suggested.
- missing the number of oocytes, ovaries used, number of uterus, etc. Provide this detailed information
Answer: Thank you for suggesting. The information of the number of COCs used in the study was added in the third line of the “2.2. Oocyte Recovery and IVM” section. However, as previously explained, all the work was carried out with material collected at the slaughterhouse. As the slaughterhouse ovary puncture technique for obtaining bovine oocytes is well described and accepted, it was not the objective of the present study to evaluate the oocyte recovery rate. Therefore, we did not record the number of ovaries.
- ln 174. use the appropriate degree symbol
Answer: Checked.
- ln 199. just one bull?
Answer: Only one tested and proved bull was used to mitigate variance. The same bull was used for all the groups. Therefore, all the found differences were due to the treatments.
- Table 2. The statistical procedure it is not adequate. Repeated measurements over time are needed; you are comparing initial blastocyst vs. blastocyst vs. expanded blastocyst. If you are just measuring the aumount of each at day7 and comparing the stage, yes the statistical procedure is correct, but explain previously clear.
Answer: Yes, we only measured the amount of each stage at day 7. As suggested, the statistical procedure was better detailed in the text.
- I would merge Table 3 with Table 2 showing the apoptotic cells ratio.
Answer: Thank you for suggesting. The authors decided to keep the tables separated due to the amount of information.
- Table 4. where are the superscripts? If there are no differences as said in ln 309-310), delete this sentence under the table.
Answer: As suggested, the line as deleted.
- figure 1. PTGS2 and HAS2, missing "=" after p value
Answer: The figure was edited, and suggestions were accepted.
- ln 334-339. why are you using capital letters? I think it is not necessary
Answer: The text was changed as suggested.
- figure 3. all superscripts in the same format
Answer: The information regarding the superscript is stated in the figure legend. The difference in superscript letters is due to the statistical analyses. Capital letters indicates tendency when the “p” value is between 0.1 and 0.05. Small letters indicate statistical difference when the “p” value is ≤ 0.05.
- Conclusions were refurmulated.
Round 2
Reviewer 3 Report
Author accomplished my previous suggestions. I congratulate them for improving the manuscript this way.